# Valproic Acid Suppresses Autoimmune Recurrence and Allograft Rejection in Islet Transplantation through Induction of the Differentiation of Regulatory T Cells and Can Be Used in Cell Therapy for Type 1 Diabetes

**DOI:** 10.3390/ph14050475

**Published:** 2021-05-17

**Authors:** Jeng-Rong Lin, Shing-Hwa Huang, Chih-Hsiung Wu, Yuan-Wu Chen, Zhi-Jie Hong, Chia-Pi Cheng, Huey-Kang Sytwu, Gu-Jiun Lin

**Affiliations:** 1Graduate Institute of Life Sciences, National Defense Medical Center, Taipei 11490, Taiwan; p8052@hotmail.com; 2Department of Biology and Anatomy, National Defense Medical Center, Taipei 11490, Taiwan; h610129@gmail.com (S.-H.H.); ph870317@mail.ndmctsgh.edu.tw (C.-P.C.); 3Department of General Surgery, En Chu Kong Hospital, New Taipei 23741, Taiwan; 01538@km.eck.org.tw; 4School of Dentistry, National Defense Medical Center, Taipei 11490, Taiwan; h6183@yahoo.com.tw; 5Department of Oral and Maxillofacial Surgery, Tri-Service General Hospital, National Defense Medical Center, Taipei 11490, Taiwan; 6Department of General Surgery, Tri-Service General Hospital, National Defense Medical Center, Taipei 11490, Taiwan; lgf670822@office365.ndmctsgh.edu.tw; 7National Institute of Infectious Diseases and Vaccinology, National Health Research Institutes, Zhunan 35053, Taiwan; sytwu@ndmctsgh.edu.tw; 8Department of Microbiology and Immunology, National Defense Medical Center, Taipei 11490, Taiwan

**Keywords:** type 1 diabetes, Valproic acid, islet transplantation, regulatory T cells, cell therapy

## Abstract

Type 1 diabetes mellitus (T1D) results from the destruction of insulin-producing β cells in the islet of the pancreas by lymphocytes. Non-obese diabetic (NOD) mouse is an animal model frequently used for this disease. It has been considered that T1D is a T cell-mediated autoimmune disease. Both CD4+ and CD8+ T cells are highly responsible for the destruction of β cells within the pancreatic islets of Langerhans. Previous studies have revealed that regulatory T (Treg) cells play a critical role in the homeostasis of the immune system as well as immune tolerance to autoantigens, thereby preventing autoimmunity. Valproic acid (VPA), a branched short-chain fatty acid, is widely used as an antiepileptic drug and a mood stabilizer. Previous reports have demonstrated that VPA treatment decreases the incidence and severity of collagen-induced arthritis and experimental autoimmune neuritis by increasing the population of Treg cells in these mouse disease models. Given the effect of VPA in the induction of Treg cells’ population, we evaluated the therapeutic potential and the protective mechanism of VPA treatment in the suppression of graft autoimmune rejection and immune recurrence in syngeneic or allogenic islet transplantation mouse models. In our study, we found that the treatment of VPA increased the expression of forkhead box P3 (FOXP3), which is a critical transcription factor that controls Treg cells’ development and function. Our data revealed that 400 mg/kg VPA treatment in recipients effectively prolonged the survival of syngeneic and allogenic islet grafts. The percentage of Treg cells in splenocytes increased in VPA-treated recipients. We also proved that adoptive transfer of VPA-induced Tregs to the transplanted recipients effectively prolonged the survival of islet grafts. The results of this study provide evidence of the therapeutic potential and the underlying mechanism of VPA treatment in syngeneic islet transplantation for T1D. It also provides experimental evidence for cell therapy by adoptive transferring of in vitro VPA-induced Tregs for the suppression of autoimmune recurrence.

## 1. Introduction

Type 1 diabetes (T1D), also known juvenile-onset diabetes or childhood-onset diabetes, is a T cell-mediated autoimmune disease resulting in insulin-producing beta cell destruction in the islet [1,2]. There are several factors involved in the pathogenesis of T1D, such as the host’s genetic background [3], the imbalance of the immune system [4], and the influence of the environment or viral infection [5]. However, the precise mechanism for this disease remains unknown. The non-obese diabetic (NOD) mouse is a frequently used mouse model for autoimmune type 1 diabetes [6,7]. This mouse model spontaneously develops autoimmune diabetes and elicits immune response in various organs, including salivary, lacrimal, thyroid, parathyroid, adrenal, testis, large bowel, and red blood cells [6,8]. Recently, regulatory T cells (Tregs) have been demonstrated to possesses an immunomodulatory effect on the immunomodulation of autoimmune diabetes in NOD mice [9]. Initially, they were described as a population of CD4^+^CD25^+^ T cells that could suppress autoimmunity by inducing immune tolerance [10]. The transcription factor forkhead box P3 (Foxp3) was also identified as a marker for Tregs, and mutation or deficiency in Foxp3 can lead to a fatal multi-organ autoimmune disease [9,11,12]. In a clinical trial for T1D, the FOXP3^+^ Tregs present in the peripheral blood were isolated by FACS and transferred into fourteen adult subjects with T1D [13]. It was found that Tregs in the recipients showed long-term retention of the phenotype and C-peptide levels persisted for 2 years [13]. In the future, T1D vaccine development could induce Foxp3^+^ Tregs to prevent islet autoimmunity in children with a high risk of T1D [14].

Valproic acid (VPA) is a frequently used drug for the treatment of epilepsy [15,16]. It was first synthesized in 1882 by Beverly S. Burton. It is naturally extracted from *Valeriana officinalis* [17]. VPA was first identified as an antiepileptic drug in 1963, after which it became a commonly used drug for epilepsy [16]. It is a short-chain branched fatty acid and acts as a potent inhibitor for histone deacetylase (HDAC) enzymes [18]. In past studies, VPA has been reported to exhibit effective anti-breast cancer and anti-inflammatory effects [19,20]. Recently, two studies showed that VPA treatment reduces spinal cord inflammation and demyelination in an experimental autoimmune encephalomyelitis mouse model by inducing apoptosis in activated T cells through the caspase-8/caspase-3 pathway [21,22]. Another study reported that VPA treatment significantly decreased disease incidence and severity in a collagen-induced arthritis (CIA) mouse model by increasing the suppressive function and number of Tregs in rheumatoid arthritis mouse model in vivo [23].

The standard therapeutic strategy for T1D patients is primarily by injection of insulin into patients to maintain a normal level of blood glucose. However, this method cannot precisely and quickly control blood glucose, and direct insulin injection, subsequently, leads to clinical complications, such as retinopathy, nephropathy, neuropathy, and macrovascular disease [24,25,26]. Using newly developed surgical procedures, islet transplantation has been reported as an effective strategy for T1D therapy and achieves moment-to-moment control of blood glucose [27]. However, it suffers from two major problems: (1) allogeneic graft rejection and (2) autoimmune recurrence [28,29]. Moreover, previous studies have reported that human islets from genetically identical twins or cadaver donors were destroyed by recurrent autoimmunity [30,31]. In this study, we investigated the therapeutic potential and underlying mechanism of VPA treatment in syngeneic islet and allogenic transplantation for T1D. Furthermore, we also explored the therapeutic potential of in vitro VPA-induced Treg cells’ differentiation for cell therapy by the adoptive transfer of these Treg cells in islet transplantation.

## 2. Results

### 2.1. VPA Treatment Prolonged the Survival of Islet Grafts after Islet Transplantation

To evaluate the protective effect of VPA in islet transplantation, we performed syngeneic and allogenic islet transplantations to monitor islet graft survival in recipients treated with 400 mg/kg VPA (Figure 1A,B). In the syngeneic transplantation, we isolated islets from 6~8-week old male NOD mice and transplanted them into the kidney sub-capsule space of diabetic female NOD recipient mice with blood glucose levels between 300 and 500 mg/dL. The mean islet graft survival time in the VPA-treated group was significantly prolonged compared with the PBS-treated group (Figure 2A). Individual islet graft survival time is presented in Table 1. In the allogenic transplantation, the islet of female C57BL/6 recipients were destroyed to induce diabetes by streptozotocin (STZ). The results showed that VPA treatment also significantly prolonged islet graft survival in allogenic islet transplantation (Figure 2B). Individual islet graft survival time is presented in Table 2. Histological assay indicated that islet grafts were destroyed at day 10 in the kidney subcapsular space of PBS-treated NOD recipients (Figure 2C). In contrast, the islet grafts were presented in the VPA-treated NOD recipients (Figure 2D). There was no secretion of insulin in the PBS-treated group (Figure 2E). In contrast, a presence of insulin secretion was still observed in VPA-treated recipients (Figure 2F). These data indicate that VPA treatment exhibits a protective effect for islet grafts in either syngeneic or allogenic islet transplantation.

### 2.2. VPA Treatment Did Not Reverse the Phenotype of Hyperglycemia in Diabetic NOD Mice

Many side effects related to VPA have also previously been reported. The most common side effects of VPA treatment are weight gain and pancreatitis [32,33]. To investigate whether VPA treatment exhibits a profound effect on blood glucose modulation in diabetic NOD mice, we treated 8~12 week-old NOD mice with different concentrations of VPA by subcutaneous injection. The results of the following analyses indicate that body weight had no significant increase in PBS and VPA (200 mg/kg, 400 mg/kg) treatment (Figure 3A). For blood glucose levels, the data showed that they were not affected by VPA treatment (Figure 3B). Since VPA-induced pancreatitis has been reported, we wanted to know if VPA would affect blood glucose metabolism. The IPGTT was used to evaluate the basal metabolism of plasma glucose concentration, and VPA treatment did not affect glucose metabolism (Figure 3C,D). Consistent with our data, the dosage of VPA treatment did not influence body weight and blood glucose of NOD mice.

### 2.3. VPA Increased IL-4-Producing CD4 T Cell and Treg Cell Population in NOD Recipients

To investigate the modulatory effect of VPA treatment on the immune system, we analyzed the proportion of immune cell subsets in the splenocyte of the recipients at day 8. There was no significant difference in the early activation marker CD69 in the CD4^+^, CD8^+^ T cells or B220^+^ B cells (Figure 4A,B). We further studied the subpopulations of CD4 T lymphocyte subsets. Although there was no significant difference in the population of Th1 cells (Figure 5A), a significant increase was observed in the percentage of Th2 cells in the VPA-treated recipients (Figure 5B). There was also no significant difference in the percentage of IL-10-producing T cells and Th17 cells (Figure 5C,D). In previous studies, it was reported that VPA treatment increased the percentage of Treg cells [10,34]. We also found that the expression of Foxp3, a critical transcription factor for Treg cells, was significantly increased by VPA treatment (Figure 6). These results indicated that the improvement of islet graft survival by VPA treatment was attributed to the increase in the population of Th2 cells and Treg cells in the VPA-treated NOD recipients.

### 2.4. VPA Treatment Induced Treg Differentiation from Naive CD4 T Cells

To investigate whether the protective effect of VPA treatment is through the increasing of Treg cell population, we first examined the proliferative capacity of splenocytes in the grafted recipients with VPA treatment. In the islet-transplanted NOD mice that were treated with VPA (at 400 mg/kg/day for 9 days) or PBS for 9 days, the proliferation of splenocytes harvested from the VPA-treated mice showed no significant increase compared to the PBS group (Figure 7A). Next, we observed the apoptosis of splenocytes in the recipients treated with VPA or PBS for 9 days. There was also no significant increase in the VPA-treated group (Figure 7B).

To investigate whether the protective effect of VPA is through the modulation of Treg cells, we examined the ability of VPA in the induction of Treg cells differentiation from naive CD4 T cells in NOD recipients. Naive CD4 T cells were sorted from 6~8week-old NOD mice and cultured in different concentrations of VPA-containing medium (0 mM, 1 mM, 2 mM, and 4 mM) with different time periods (0, 12, and 24 h). The percentages and absolute numbers of Treg cells were significantly increased following VPA treatment in a dose-dependent manner (Figure 8A,B). The results suggest that the protective effect of VPA could be attributed to the VPA-induced Treg cells’ differentiation in the syngeneic or allogeneic islet transplantations.

### 2.5. VPA Induced Treg Cell Differentiation from Naive CD4 T Cells by Increasing the Expression of the Transcription Factor STAT5 and Histone3(H3) Acetylation

In past studies, the activation of STAT5 transcription activity was an important factor for the differentiation of inducing Treg cells by Foxp3 expression [35,36]. To investigate the mechanism of the VPA-induced Treg cell differentiation from the naive CD4 T lymphocytes, we examined the influence of the VPA treatment on the activation of STAT5. The expression of phosphorylated STAT5 (p-STAT5) in the splenocytes of the VPA-treated group significantly increased compared to the PBS-treated group (Figure 9A,B).

As previously described valproic acid is a histone deacetylase inhibitor. VPA regulates the conformation of chromatin and affects the gene transcription. Therefore, we investigated whether VPA exhibits an epigenetic effect in the CD4 T cells. These results suggest that the effect of VPA in the induction of Treg cell differentiation could be via an epigenetic effect by increasing the acetylation of H3 (Figure 9C,D). Our results showed that the acetylation of H3 in CD4 T cells significantly increased under VPA treatment.

### 2.6. Adoptive Transfer of In Vitro VPA-Induced Regulatory T Cells Prolonged Islet Graft Survival

Naive CD4 T cells harvested from the NOD mice were cultured for 24 h with IL-2 (5 ng/mL) and then treated with PBS or 2 mM VPA for 24 h in vitro. We adoptively transferred 1 × 10^6^ differentiated Treg cells at day 1 and day 3 after islet transplantation (Figure 10A). There was a significant prolonging islet survival in the VPA-treated cell transferred groups compared to the PBS-treated cells group (Figure 10B). Histological and immunohistochemical assays showed more intact islet grafts (Figure 10C,D) and more insulin producing islets in VPA-treated cell transferred recipients (Figure 10E,F). Individual islet graft survival time is presented in Table 3. These results indicated that adoptive transfer of VPA-induced Treg cells to NOD recipients exhibited a protective effect for islet grafts. Our data showed that in vitro VPA-treated cell adoptive transfer therapy also exhibits a protective ability to in vivo VPA treatment, suggesting a clinical therapeutic potential by cell therapy in islet transplantation for type 1 diabetes patients.

## 3. Discussion

Whole-pancreas transplantation was previously used in the treatment of T1D; however, it leads to many complications including technical failures (thrombosis, bleeding, leak) with a rate of 6.5%, acute rejection in 12% of cases, and CMV infection with a rate of 10% [37]. Islet transplantation can successfully recover the function of β cells by injection of islet grafts via a recipient’s liver portal vein [27]. However, the islet graft will suffer from immune attacks including autoimmune recurrence and allograft rejection. Therefore, it is an important issue to overcome these immune responses in islet transplantation. Our data indicate that VPA could be successful in protecting islet grafts and prolonging graft survival in islet transplantation.

A previous study demonstrated that VPA attenuates inflammation in experimental autoimmune neuritis and suppresses mRNA levels of inflammatory cytokine [38], VPA treatment also increases the population of Foxp3^+^ Treg cells; in contrast, the population of IL-17^+^ T cells decreased. In a previous murine heart transplantation study, VPA also significantly inhibited B cell class swish recombination and plasma cell differentiation, thereby reducing the levels of donor-specific antibody in VPA-treated mice [39]. In our study, 400 mg/kg VPA treatment by subcutaneous injection effectively suppressed recurrent autoimmunity and significantly prolonged islet graft survival in NOD recipients. Moreover, we also examined whether VPA treatment elicits a toxic effect, influencing the secretion of insulin. Our data indicate that VPA treatment at the dosage of 400 mg/kg in NOD mice did not affect body weight and insulin secretion, suggesting that the protective effect of VPA in NOD recipients was not due to the regulation of blood glucose metabolism or the elevation of insulin production.

In T1D, Th1 cells play a pathogenic role for the initiation of the disease process [40]. In contrast, administration of Th2 cytokine, IL-4, can prohibit the onset of autoimmune diabetes [41]. IL-4 is also a negative regulator that inhibits the differentiation of naive CD4^+^ T cell into Th1 or Th17 cells [42,43]. Moreover, IL-4 can induce IL-10, inducing Foxp3^+^ cell migration to the injury site and which has an anti-inflammatory effect. In our analysis of the population of lymphocytes after VPA treatment in the islet transplantation model, we found that VPA treatment did not affect T cell and B cell activation but significantly increased the population of Th2 cells. In a previous study, VPA induced apoptosis in activated T cells and maintained the immune homeostasis [21]. However, our data showed that VPA treatment did not induce apoptosis in activated T cells of NOD mice. In a previous study, in the experimental coxsackievirus-B3-induced myocarditis model, VPA treatment downregulated the expression of IL-17A and upregulated IL-10 cytokine of infected mice through the induction of Treg cell differentiation [44]. In this study, we also found that VPA treatment increased Treg cell’s population. We further observed that VPA induced the differentiation of naive CD4 T cells and Treg cells. Next, we further investigated which signaling pathway was affected by VPA for the induction of the differentiation of Treg cells. A previous study reported that the STAT5 signaling pathway contributes to the expression of FOXP3 in CD4^+^ CD25^+^ Treg cells [35]. Our results showed that the phosphorylation of STAT5 increased under VPA treatment. Epigenetic gene regulation also contributes to the development of many diseases, such as type 1 diabetes, and acetylation of histones plays an important role [45]. VPA has been found to act as an HDAC inhibitor and impacts on chromatin remodeling through regulating histone deacetylases (HDACs) and histone acetyltransferases (HATs) [46]. The transcription factor STAT5 interacts with p300 that acetylates histone H3 at lysine 27 (H3K27ac) promoting transcription [47]. Our results found that phosphorylation of STAT5 and acetyl-H3 increased after VPA treatment, suggesting that VPA promotes differentiation of Treg cells by inducing the activation of the STAT5 signaling pathway and by suppressing histone H3 deacetylation.

Although in vivo VPA treatment effectively prolonged the survival of syngeneic or allogeneic islet grafts, VPA treatment is associated with many adverse side effects, such as hepatotoxicity, hyperammonemia, and insulin resistance [48]. The clinical trial also showed that VPA enhances body weight gain at six months in patients after therapy with childhood epilepsy [41]. A recent animal study demonstrated that long-term treatment epileptic WAG/Rij rats increases the level of serum alanine aminotransferase and aspartate aminotransferase [49]. In the clinical observations, long-term VPA therapy in the treatment of epilepsy is associated with reversible or irreversible hepatotoxicity [50]. To avoid these adverse side effects, we investigated whether adoptive transfer in vitro of VPA-induced Treg cells to the recipients can exhibit similar protective effect as in vivo VPA treatment. Surprisingly, this treatment exhibited a similar protective effect in NOD islet grafts for the in vivo VPA treatment. This result indicates that adoptively transferred VPA-induced Treg cells can be used as a cell therapeutic strategy to effectively prolong islet grafts’ survival and avoid the adverse effects of the in vivo VPA treatment. Furthermore, a better protective efficiency than twice adoptive transfers in our experiment could be achieved by increasing the time or the number of VPA-treated cell to maintain a long-term immune modulatory effect.

## 4. Materials and Methods

### 4.1. Animal Model

The NOD/ShiLtJ strain is a polygenic model for autoimmune type 1 diabetes and NOD/scid mice were purchased from Jackson Laboratory (Bar Harbor, ME, USA) and subsequently bred at the animal center of the National Defense Medical Center in Taipei, Taiwan, under specific pathogen-free conditions. Balb/c and C57BL/6 mice were purchased from the National Laboratory Animal Center in Taipei, Taiwan.

### 4.2. Islet Isolation and Transplantation

Diabetic NOD females with blood glucose concentrations 3–5 mg/mL for two consecutive days were selected as recipients, and NOD male mice aged 5–8 weeks were used as islet donors. Islets were purified from 6-week-old male NOD mouse using the collagenase-digesting method as described previously [51]. Collagenase buffer was prepared with Hank’s balanced salt solution containing 1.5 mg/mL collagenase (Sigma-Aldrich, St Louis, MO, USA) and injected into the pancreas via a common bile duct. The pancreas was digested in a 37 °C water bath for 20 min and then the islets were separated using a density gradient using a Histopaque 1077–1 (Sigma-Aldrich). Islets with a diameter between 75 μm and 250 μm were handpicked using a dissecting microscope. Finally, we collected a total of approximately 650 islets that were implanted into the left renal capsule of newly diabetic NOD female mouse whose blood glucose concentrations were 3–5 mg/mL. For allogenic transplantation, islets were purified from 6-week-old male Balb/c mouse and transplanted into streptozotocin (STZ)-induced diabetic C57BL/6 mouse.

### 4.3. Blood Glucose Monitoring

Blood glucose concentration was monitored daily after islet transplantation. Graft rejection and loss of function was defined as blood glucose levels higher than 3 mg/mL for two consecutive days.

### 4.4. Naive T Cell Sorting

Naive CD4^+^ T cells were harvested and sorted from the spleen of NOD mouse by cell separation magnetic beads. The BD IMagTM Mouse CD4 T Lymphocyte Enrichment set-DM (BD Biosciences, San Jose, CA, USA) was used for negative selection of CD4 T lymphocyte by removing non-CD4 T cells from splenocytes. Cells were re-suspended at a concentration of 1 × 10^6^ cells/mL in RPMI1640 medium supplemented with 10% fetal bovine serum and 1% penicillin and streptomycin. For the isolation of naive CD4 T cells, the selected CD4 T cells were added with biotinylated mouse CD4 T lymphocyte antibody and 2 μL biotinylated anti-CD25 antibody, and then incubated for 15 min. After washed with medium, the streptavidin particles were added 5μL into the tube for 30 min in 4 °C refrigerator. The cells were then put on the cell separation magnetic platform for 8 min, repeated three times. These positive fractions of the cells were isolated as naive CD4 T cell.

### 4.5. In Vitro of Treg Cell Differentiation

Naive CD4^+^ T cells were harvested and sorted from the splenocytes of male NOD mice and then cultured for 1 day with human IL-2 cytokine at 5 ng/mL. The next day, the naive CD4^+^ T cells were cultured with PBS, or 1 mM, 2 mM, or 4 mM VPA for 0, 12, and 24 h, respectively. CD4^+^CD25^+^Foxp3^+^ cells were then measured from these PBS- or VPA-treated naive CD4^+^ T cells by flow cytometry.

### 4.6. T Cell Apoptosis Analysis

The CD4^+^ T cells were harvested and isolated from the splenocytes of NOD mouse in complete RPMI 1640 medium containing 5 ng/mL IL-2 for 1 day at a density of 1 × 10^6^ cell. Then, these cells were cultured with different concentrations of VPA in complete RPMI 1640. The cells were stained with propidium iodide (PI) and fluorescein isothiocyanate (FITC)-annexin V in combination with anti-CD4 or anti-CD25 antibody. The percentage of apoptotic cells was determined by flow cytometry.

### 4.7. Flow Cytometry

Lymphocytes were harvested from spleen and 1 × 10^6^ cells were stained with 2 μg/mL allophycocyanin (APC) conjugated anti-mouse CD4 (clone GK1.5), phycoerythrin (PE)-conjugated anti-mouse CD69 (clon H1.2F3), peridinin chlorophyll protein complex (PerCP-Cy5.5)-conjugatedand anti-mouse CD8α (clone 53-6.7), and 5 μg/mL FITC-conjugated anti-mouse B220 (clone RA3-6B2) in 100 μL of flow buffer for 30 min at 4 °C. For Foxp3 staining, the 1 × 10^6^ cells were first stained with 2 μg/mL antibodies to surface CD4 (APC conjugated), CD8 (PerCP-Cy5.5-conjugated), and PE-conjugated anti-mouse CD25 (clone PC61) in 100 μL of flow buffer for 30 min at 4 °C, and then fixed and permeabilized overnight with 1 mL Fixation/Permeabilization working solution (eBioscience Inc., San Diego, CA, USA). After fixation and permeabilization, the cells were stained with 5 μg/mL FITC-conjugated anti-Foxp3 (clone FJK-16S) (eBioscience Inc.) in 100 μL permeabilization buffer. For intracellular cytokine staining, the cells were stimulated for 4–6 h with 20 ng/mL phorbol 12-myristate 13-acetate (PMA), 1 μM ionomycin, and 4 μM monensin. The 1 × 10^6^ stimulated cells were stained with 2 μg/mL antibody to surface CD4-APC, and CD8-PerCP-Cy5.5 100 μL of flow buffer on ice for 25–30 min (in the dark), and washed with 1 mL of FACS buffer (PBS containing 0.5% FBS). After being washed, the cells were fixed overnight with 0.2 mL of IC Fixation Buffer (eBioscience Inc.). The cells were then stained with 5 μg/mL anti-mouse IL-10-FITC, IFN-γ -FITC, IL-4-PE, and IL-17-PE-conjugated antibodies on ice for 30 min in 100 μL permeabilization buffer. Flowcytometric analysis was performed with a FACS Calibur (BD Pharmingen) and CellQuest software (Becton Dickinson, San Jose, CA, USA).

### 4.8. Low-Dose Streptozotocin(STZ) Induction Diabetes

This protocol is used by DiaComp members to induce diabetes in B6 mice by STZ (Sigma-Aldrich, Saint Louis, MO, USA). The STZ was dissolved into Na-Citrate buffer at a final concentration of 7.5 mg/mL. Mice received intraperitoneal (i.p.) injection of STZ solution at 50 mg/kg once per day for 5 days consecutively. The diabetic mice were determined via blood glucose level higher than 3 mg/mL.

### 4.9. Histological and Immunohistochemical Assays

Kidneys transplanted with islets were harvested from NOD recipients and then embedded in paraffin. Sections (5 μm in thickness) were cut and stained with hematoxylin and eosin (H&E) staining and then observed with light microscopy. In the immunohistochemical assay, the kidney section slides were stained with antibodies against insulin (abcam, Cambridge, MA, USA, ab7842) over night and then stained with second anti-guinea pig IgG (Bethyl, A60-110p) antibodies for 1hr. Finally, the slides were stained with hematoxylin and analyzed via light microscopy.

### 4.10. Adoptive Transfer of Regulatory T Cells

Naive CD4 T cells harvested from the NOD mice were cultured for 24 h with IL-2 cytokine (5 ng/mL) and then treated with 2mM VPA for 24 h in vitro. The 1 × 10^6^ differentiated Treg cells were adoptively transferred into the diabetic NOD mice after islet transplantation by intraperitoneal (i.p.) injection. This protocol was followed according to our previous study [52].

### 4.11. Protein Extraction and Western Blot

The protein samples were extracted from the splenocytes of the NOD mice that were treated with PBS or VPA at 1 mM, 2 mM or 4 mM using the PROPREP™ Protein Extraction Solution (iNtRON Biotechnology, Gyeonggido, Korea). A sample in the protein extraction solution was homogenized by an Ultrasonic Homogenizer (Misonix, Farmingdale, NY, USA), and it was then incubated on ice for 20–30 min to lyse the cells. After centrifugation at 13,000 rpm for 10 min at 4 °C, the supernatant was transferred to a new Eppendorf tube. A total of 10 μg of the protein sample was separated on 10% SDS-PAGE, and then transferred to a PVDF membrane (Millipore, Billerica, MA, USA). The membrane was blocked with 5% skim milk at room temperature for 1 h and was then incubated in a buffer with a rabbit anti-STAT5 antibody, anti-acetylated H3, or a mouse anti-β-actin antibody overnight. After washing with PBST (0.05% Tween20 in PBS) three times, the membrane was incubated in the hybridization buffer with a horseradish peroxidase (HRP)-conjugated goat anti-rabbit IgG antibody (1:2000; Santa Cruz Biotechnology, Inc., Dallas, TX, USA) or an HRP-conjugated goat anti-mouse IgG antibody for 1 h. The membrane is subsequently washed with PBST three times. After incubation with the chemiluminescent HRP substrate (Millipore), the signals were detected by the LAS-3000 imaging system (Fujifilm, Tokyo, Japan).

### 4.12. Statistical Analysis

GraphPad Prism 8 was used as statistical software in this study. The data are presented as the mean ± SD or SEM. The significance of islet graft survival time between the PBS-treated and VPA-treated groups was determined via Kaplan–Meier survival analysis. The significance of diabetic frequency between the PBS-treated and VPA-treated NOD mice was also determined via Kaplan–Meier survival analysis. For the remaining experiments, *p*-values were calculated using the two-tailed Student’s t test or ANOVA test. Differences were considered significant at *p* < 0.05.

## 5. Conclusions

Our study demonstrated that VPA treatment effectively prolonged the survival of syngeneic and allogeneic islet grafts in diabetic NOD and C57BL/6 mouse models, respectively. Its immune suppressive effect is via the induction of the differentiation of Treg cells from the naïve CD4 T cells by increasing the phosphorylation of STAT5 as well as by suppressing the deacetylation of H3. Our results also first demonstrated that in vitro induction of Treg cells from sorted naïve CD4 T cells by VPA can be used in the cell therapy for the treatment of T1D patients in order to avoid the toxicity of in vivo VPA treatment.

## Figures and Tables

**Figure 1 pharmaceuticals-14-00475-f001:**
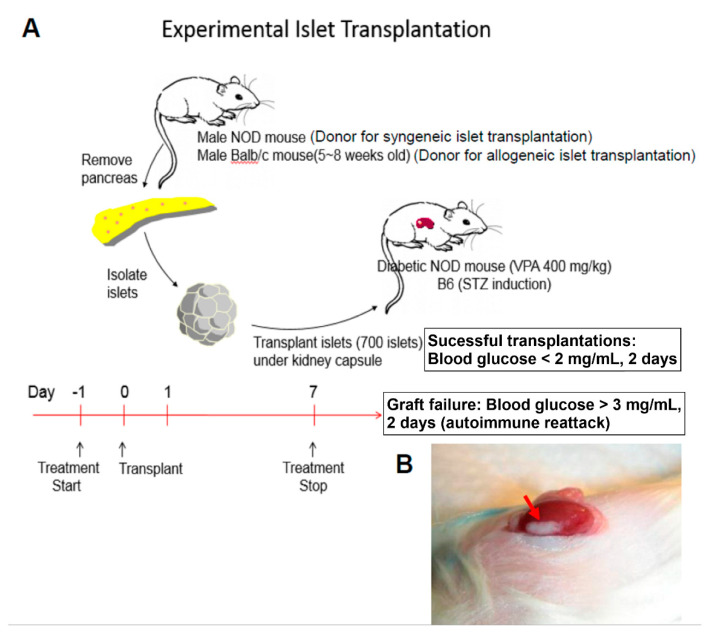
Syngeneic or allogeneic islet transplantation were performed to treat diabetic recipients. (**A**) Newly isolated islet from male NOD or Balb/c mice were transplanted into the kidney subcapsular space of newly diabetic NOD or streptozotocin (STZ)−induced diabetic C57BL/6 recipients, respectively. Valproic acid (VPA) treatment prolonged islet graft survival in the syngeneic and allogeneic islet transplantation of non−obese diabetic (NOD) or C57BL/6 mice. The recipients were treated with nine dosages of VPA 400 mg/kg. (**B**) The red arrow indicates the presence transplanted islet graft in the subcapsular space of NOD recipients’ kidney.

**Figure 2 pharmaceuticals-14-00475-f002:**
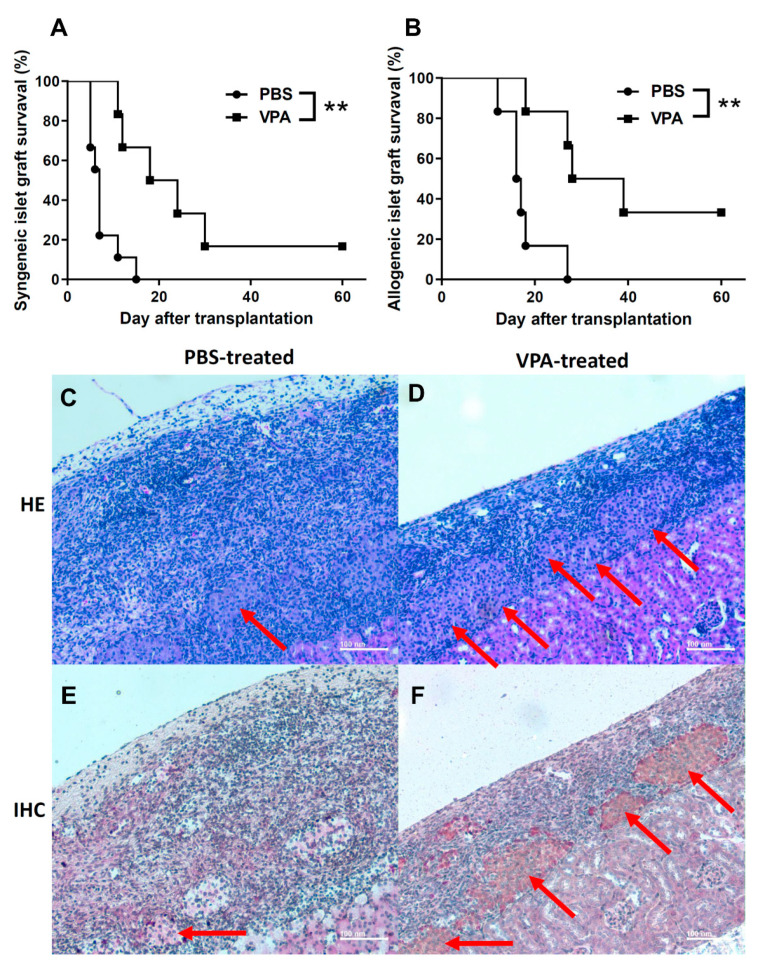
The effect of VPA treatment in the syngeneic and allogeneic islet transplantation of non-obese diabetic (NOD) and B6 mice. (**A**) The survival time of Syngeneic islet grafts was significant longer in VPA-treated group (*n* = 6) compared to the PBS-treated group (*n* = 9, ** = *p* < 0.001). The survival time of islet grafts in VPA-treated mice was significantly prolonged to 63 days (*n* = 6, ** = *p* < 0.001 compared to the PBS-treated groups). (**B**) In the allogeneic islet transplantation, the survival of islet grafts in the streptozotocin (STZ)-induced diabetic C57BL/6 was significantly longer in the VPA treat-ed group compared to the PBS-treated controls (*n* = 6). Histological assay indicated that islet grafts were destructed at day 10 in the kidney subcapsular space of (**C**) PBS-treated NOD recipients. In contrast, (**D**) the islet grafts were presented in the VPA-treated NOD recipients. The secretion of insulin was assessed by immunohistochemical staining. Red arrow indicates the insulin staining in the (**E**) PBS-treated group and the (**F**) VPA-treated recipients.

**Figure 3 pharmaceuticals-14-00475-f003:**
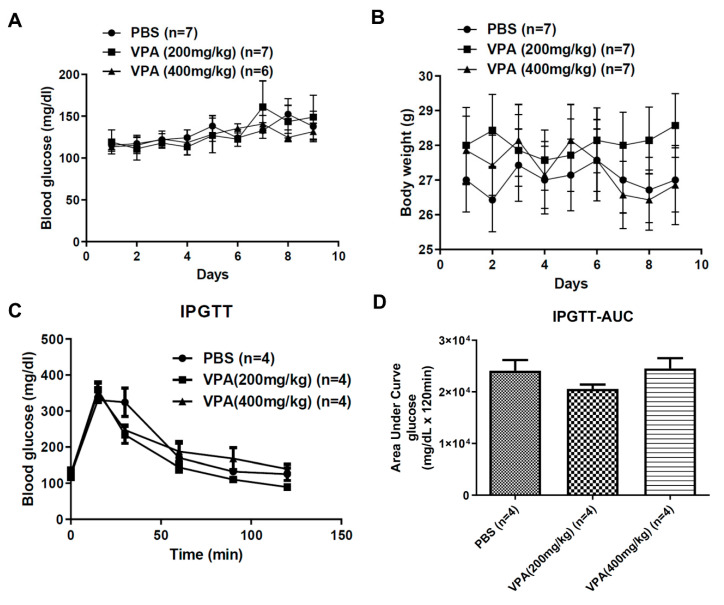
The effect of VPA treatment in the body weight and insulin secretion of NOD recipients. (**A**) There were no significant differences in the body weight and (**B**) blood glucose between PBS-treated and VPA-treated mice (*p* > 0.05, *n* = 7). (**C**) IPGTT was performed to examine the influence of VAP on the metabolism of blood glucose. Glucose was i.p. injected into NOD mice at 2 g/kg concentration following a fast for 6 h. (**D**) Area under curve (AUC) also showed no significant differences (*n* = 4).

**Figure 4 pharmaceuticals-14-00475-f004:**
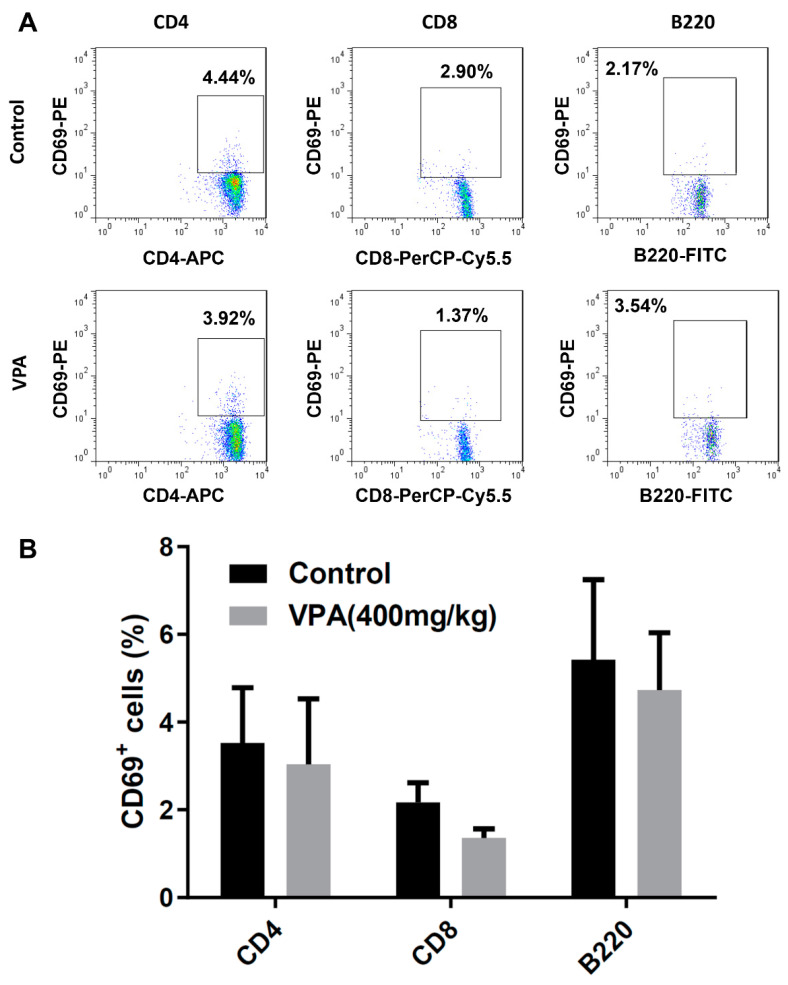
The effect of VPA treatment in the activation of lymphocyte in NOD recipients. (**A**) Representative plot of the expression of CD69 in CD4 or CD8 T cells and B cells (B220). (**B**) There was no significant difference in the percentage of CD69 in the CD4, CD8 T cells or B cells.

**Figure 5 pharmaceuticals-14-00475-f005:**
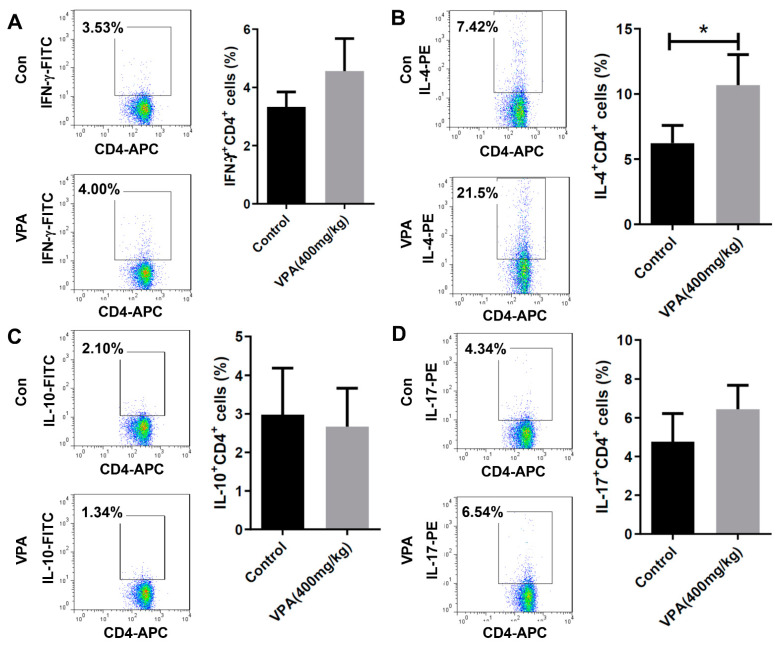
The effect of VPA treatment in the subsets of helper T cells in the spleen of NOD mouse after islet transplantation. The population of Th1, Th2, IL-10-producing and Th17 CD4 T cells in the PBS-treated or VPA-treated NOD recipients was analyzed by flow cytometry. (**A**) No significant difference in the population of Th1 cells. (**B**) A significant increase was observed in the percentage of Th2 (IL-4-producing CD4 T) cells in the VPA-treated recipients. No significant difference in the percentage of (**C**) IL-10-producing CD4 T cells and (**D**) Th17 (IL-17-producing CD4 T) cells. Data are expressed as the mean ± SEM (* = *p* < 0.05).

**Figure 6 pharmaceuticals-14-00475-f006:**
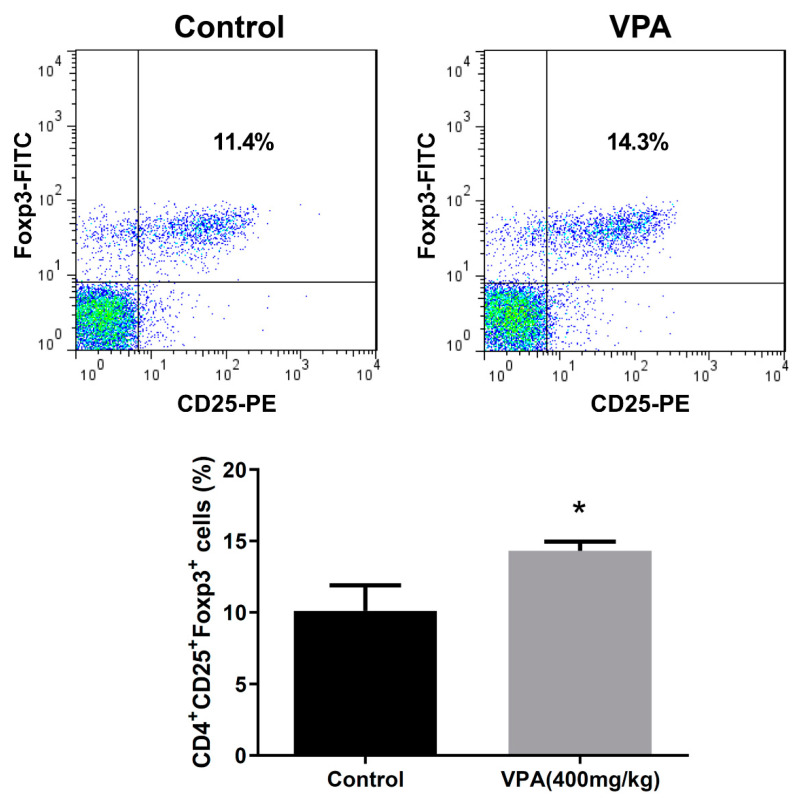
The effect of VPA treatment in the population of Treg cells in the spleen of NOD mouse after islet transplantation. The population of Treg cell in the splenocyte of PBS-treated or VPA-treated NOD recipients was analyzed by flow cytometry. The percentage of Treg cells was significant higher in VPA-treated group than PBS-treated control. Data are expressed as the mean ± SEM (* = *p* < 0.05; *n* = 4).

**Figure 7 pharmaceuticals-14-00475-f007:**
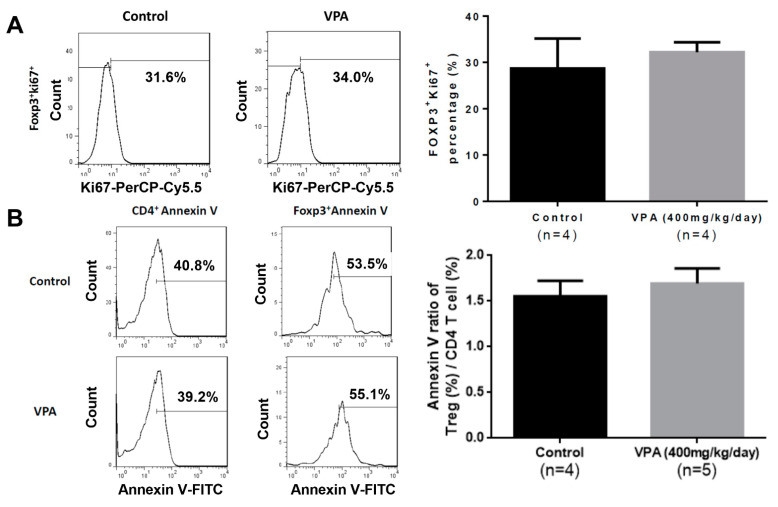
The effect of VPA treatment in the proliferation and apoptosis of Treg cells in the NOD recipients. (**A**) The proliferation of Treg cell was analyzed by the staining of proliferation marker Ki67. The percentage in the Ki67^+^ cells showed no significant difference. Data are expressed as the mean ± SEM. (**B**) The apoptosis of CD4 or Treg cells was analyzed by Annexin V staining. The ratio of Annexin V positive Treg cell and CD4 T cell showed no significant difference between control and VPA-treated group. Data are expressed as the mean ± SEM.

**Figure 8 pharmaceuticals-14-00475-f008:**
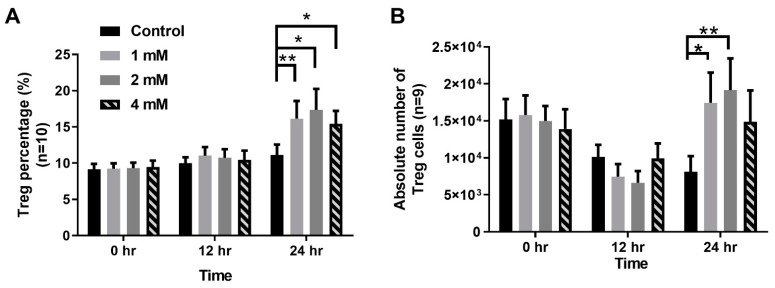
The effect of VPA treatment in the induction of Treg cells’ differentiation from the naive CD4 T cells of NOD mouse. Naive CD4 T cells were cultured with differential concentrations of VPA solutions (0 mM, 1 mM, 2 mM, and 4 mM) for different incubation times (0 h, 12 h, and 24 h). After VPA solution treatment, the percentage of lymphocytes were analyzed via flow cytometry. (**A**) The percentage of Treg cells differentiated from naïve CD4 T cells was significantly increased following VPA treatment at different concentrations of VPA at 24 h. (**B**) The absolute number of Treg cells differentiated from naïve CD4 T cells was calculated by multiplying the total number of the cells with the percentage of Treg cells. It also showed a significantly increased following VPA treatment at different concentrations of VPA at 24 h. Data are expressed as the mean ± SEM (n = 6; * *p* < 0.05, ** *p* <0.01).

**Figure 9 pharmaceuticals-14-00475-f009:**
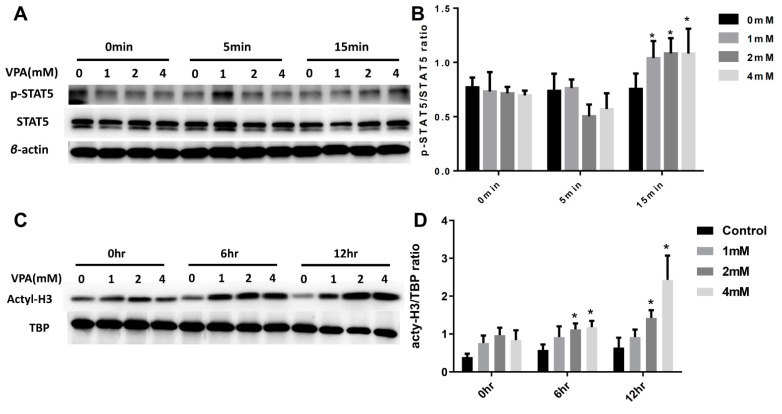
The molecular mechanism of VPA treatment in the induction of Treg cells’ differentiation from naïve CD4 T cells of NOD mouse. The CD4 naive T lymphocytes were cultured with various concentrations of VPA solutions (0 mM, 1 mM, 2 mM, and 4 mM) at different incubation times. (**A**) The levels of STAT5 and (**B**) phosphorylated STAT5 (p-STAT5) were measured by Western blot. (**C**,**D**) The levels of histone acetylation (acetyl-H3) were assessed by Western blot. Data are expressed as the mean ± SEM (* = *p* < 0.05).

**Figure 10 pharmaceuticals-14-00475-f010:**
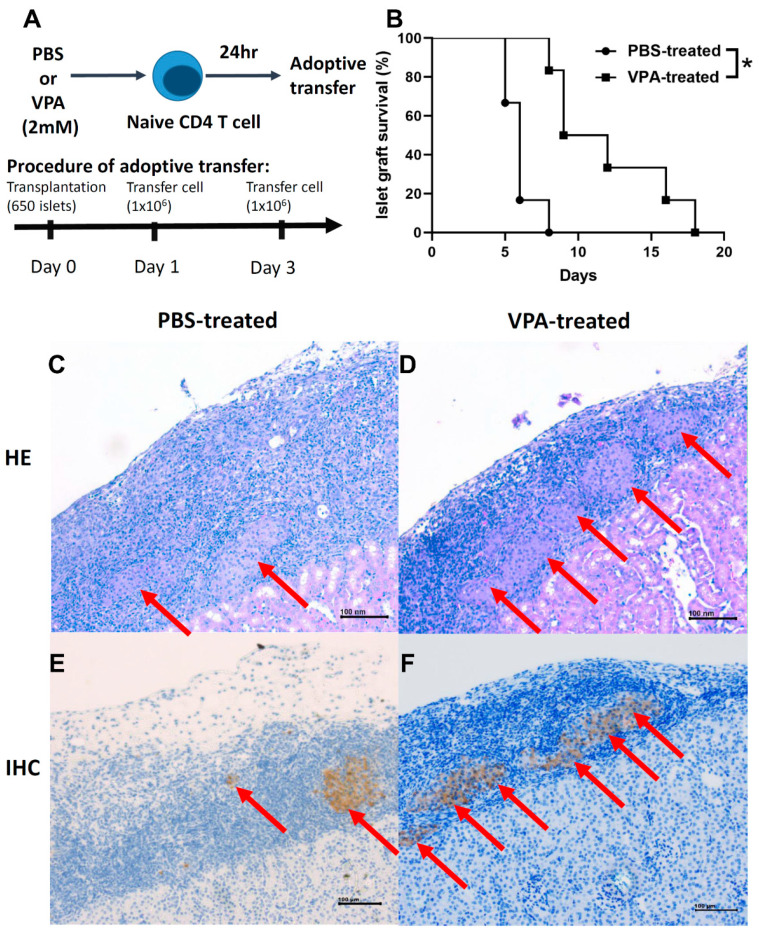
The effect of adoptively transferring VPA-induced Treg cells in the islet graft survival in syngeneic islet transplantation. (**A**) Naïve CD4 T cells were harvested from the spleen of the NOD mice. The naïve naive CD4 T cells were cultured in the medium with 2 mM VPA or PBS. A total of 1 × 10^6^ PBS-treated or VPA-treated cells were collected and adoptively transferred into the NOD mice twice at day 1 and day 3 after islet transplantation. (**B**) The islet graft survival was significant longer in the VPA-treated group compared to the PBS-treated group (* *p* < 0.05; n = 6). (**C**,**D**) The infiltration of lymphocytes in islet grafts was lower in the VPA-treated Treg transfer into NOD recipients at day 6 post-islet transplantation. In contrast, more islet grafts were presented in the VPA-treated Treg transferred NOD recipients. The secretion of insulin was assessed by immunohistochemical staining. Red arrow indicates the insulin staining in the (**E**) PBS-treated group and the (**F**) VPA-treated Treg transferred NOD recipients.

**Table 1 pharmaceuticals-14-00475-t001:** The survival time of syngeneic islet graft in the NOD recipients.

Group	Individual Graft Survival Time (Days)	Number	Average Survival Time
PBS	5, 5, 5, 6, 7, 7, 7, 11, 15	9	7.55
VPA	11, 12, 18, 24, 30, 63	6	26.33

**Table 2 pharmaceuticals-14-00475-t002:** The survival time of allogeneic islet graft in the C57BL/6 recipients.

Group	Individual Graft Survival Time (Days)	Number	Average Survival Time
PBS	12, 16, 16, 17, 18, 27	6	17.66
VPA	18, 27, 28, 39, 67, 82	6	43.5

**Table 3 pharmaceuticals-14-00475-t003:** The survival time of islet grafts in PBS- or VPA-treated cell transferred NOD recipients.

Group	Individual Graft Survival Time (Days)	Number	Average Survival Time
PBS	5, 5, 6, 6, 6, 8	6	6
VPA	8, 9, 9, 12, 16, 18	6	10.3

## Data Availability

The data presented in this study are available in the main text or on request from the corresponding author.

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
