# Peer review of "Valproic Acid Suppresses Autoimmune Recurrence and Allograft Rejection in Islet Transplantation through Induction of the Differentiation of Regulatory T Cells and Can Be Used in Cell Therapy for Type 1 Diabetes"

_pharmaceuticals, 2021, doi:10.3390/ph14050475_

Round 1
Reviewer 1 Report
The authors attempt to show the efficacy of VPA treatment for the prolongation of islets of Langerhans graft survival syngeneic and allogeneic models of transplantation. They try to show that this is due to the expansion of Treg cells. I believe that despite quite impressive in vivo effects, which are rather compelling, the efforts of the authors to pinpoint the mechanism(s) of the VPA effect were not extremely convincing. Moreover, the manuscript is filled with A LOT of non-significant data, which can be put in the supplement (e.g. Figs 3 and 4).
General comments:
- There are a few places where authors confuse plurals and singulars (e.g. Keywords “Regulatory T cell”, line 380 “sorted from the spleen of NOD mice)
- Usage of concentration /dl may be confusing as it’s a rarely used unit, I would suggest /ml
- Flow cytometry methods section 4.7 lacks concentrations of the antibodies used, which must be fixed. The section also lacks a lot of spaces. This is yet again a very recurring theme in the manuscript as certain parts seem to have been pasted in a hurry.
- Which statistical software has been used in the manuscript?
- Figure legends should not tell the reader the data interpretation, but rather describe the figures in an objective manner: what has been done and using which methods (e.g. reader should be told that Figs 2E and F supposedly show insulin staining and not e.g. The secretion of insulin was also disappeared in (E) PBS-treated group (indicated by the red arrows) and (F) a presence of insulin secretion was observed in VPA-treated recipients (indicated by the red arrows).”).
- Same with Figure legend titles
- It is common nowadays to have a fluorophore indicated together with the marker for flow cytometric plots
- Flow cytometric plots are low quality and at least the percentages of gated populations should be shown with a much larger font. Other cosmetic things such as deletion of FL3-H:: CD8 can be done. This is especially evident in Fig 5, as those titles run into the headings of the authors.
- In many places in the manuscript gamma interferon is abbreviated to IFN-r, which should be IFN-É£
- Gate titles and percentages should not get in the way of the cell populations, but rather placed to the side, as clearly evident in Figs 5A and B.
Specific comments:
- Treg abbreviation introduced in line 34 after the first usage in line 29
- Line 41 splenocytes, not splenocyte
- Space after coma in line 64
- Fig 4A should depict CD69 on the y axes more clearly, also those plots are not density plots as stated in the Fig legend.
- Figure 6 is extremely underwhelming and not extremely believable, especially without the numbers of animals in each group.
- Figure 8 is extremely grainy and low quality and needs to be improved
- It’s never stated how the absolute numbers of Tregs were calculated in Fig 8B. Moreover, figure legend is more of a mess than usual
Reviewer 2 Report
Lin and co-workers offer an interesting study to support long-term graft of (murine) pancreatic islets by pharmacological treatment with valproic acid.
The specific effects as epigenetic modulators and regulatory effects on immune cells have been evaluated both in vivo and in vitro. Additional analysis are recommended based on such preliminary observations. However, the study is quite complete and of interest.
We recommend revision and small editing before publication. And maybe few more adds, to offer a more compelte view and solid results to the reader.
Introductory lines 75-84 can be shorten up and combined with the early onset of Treg in T1D (lines 59-61).
Was the survival of allogenic panc islets significantly augmented upon treatment with VPA (line 110)?
Figure 1: Why were all the syngenic panc islets rejected faster (day 18) compared to allogenic islets (day 26 or similar)?
Lines 145-147 are redundant with Intro and not strictly required for section 2.2
First 6 lines in Discussion are Intro redundant and not required. Consider to remove it.
Similarly, VPA description at page 12 (line 295 and more) is largely replicating what’s already stated in the Introduction. Please revise and shorten up (or completely remove such introductive part).
Finally, the authors correctly stated the side effects of long-term administration of VPA (lines 339-340: VPA treatment is associated with many adverse side effects such as hepatotoxicity, hyperammonemia, and insulin resistance). Quick immunohistological analysis and serological evaluation for hepatic transaminases and liver damages after VPA treatment would be instrumental and relevant for the current study.
Round 2
Reviewer 1 Report
The authors have improved some parts of the manuscript quite significantly. I will leave the spell checking and similar things to the journal staff.
However, there are still two major issues:
- Despite correcting the figure legend and title for Fig 2, the other figure titles and legends were left as they were (i.e. all of them interpret, rather than present the data.
- There were some improvements in the flow cytometric plots, regarding the increased fonts for the percentages of gated events. However, all of those should be out side of the gates and the FlowJo mask fonts and the repeating percentages should be deleted, together with tinted rectangles introduced by flowjo.
Finally, the amount of staining antibodies used for flow cytometry should be depicted either as a concentration or a dilution.
Reviewer 2 Report
I am fine with the changes and adds the authors applied to the manuscript. Hepatotoxic studies are relevant, but we totally understand the lack of time to conduct proper analysis, and we confide the authors will perform proper analysis in future studies
